# Acceptance of a homestay program and attitude toward community medicine among medical students

**Tsuneaki Kenzaka**[1,2]*, **Shinsuke Yahata**[3], **Ken Goda**[1,2], **Ayako Kumabe**[1], **Hozuka Akita**[2], **Masanobu Okayama**[3]

**1** Division of Community Medicine and Career Development, Kobe University Graduate School of Medicine, Kobe, Japan, **2** Department of Internal Medicine, Hyogo Prefectural Tamba Medical Center, Tamba, Japan, **3** Division of Community Medicine and Medical Education, Kobe University Graduate School of Medicine, Kobe, Japan

* smile.kenzaka@jichi.ac.jp

**Data Availability Statement:** All relevant data are within the manuscript.

**Funding:** The authors received no specific funding for this work.

## Abstract

### Background

In community-based medical education, opportunities for medical students to interact with local residents are important. To facilitate such interaction, we aimed to evaluate acceptance of a homestay program and attitude toward community medicine among medical students.

### Methods

The participants (n = 39) were allowed to stay in the local homes of residents for one night in August 2016, 2017, and 2018. Before and after the homestays, the students responded to a self-reported questionnaire using the visual analog scale (VAS; 0–100 mm). The questionnaire included four questions on homestay/community medical training and community medicine and four questions about attitude toward community medicine in the local areas of medical students. Then, we compared the VAS scores before and after training.

### Results

The VAS scores for all questions about homestay/community medical training and community medicine significantly increased: "Is it worthwhile for you to have experience in the field of community medicine," "Did you find the homestay enjoyable," "Does the homestay add educational significance to the program," and "Is direct interaction with residents meaningful?" For the two questions about attitude toward community medicine, the VAS scores significantly increased: "Is there a challenge to practicing community medicine" and "In the future, do you want to work in Tamba area where you stayed?"

### Conclusions

The medical students were extremely enthusiastic about the educational program for community medicine involving residential homestays, which improved their attitudes toward

**Competing interests:** The authors have declared that no competing interests exist.

practicing community medicine. Moreover, the students appreciated that their training sites could become their future workplaces.

## Background

The Model Core Curriculum for Medical Education in Japan (AY 2016 Revision) emphasized the importance of community-based medical training and recommended providing opportunities to learn and provide medical care to local residents [1]. The educational program for community medicine increased the awareness of medical students about community healthcare [2]. In particular, the program included health/patient education to increase the effect of practical training [2]. Particularly in community-based medical education, opportunities for medical students to interact with local residents are important [2, 3].

To facilitate such interaction, we allowed students to stay in the homes of local residents for one night in August 2016, 2017, and 2018. A limited number of medical institutions incorporate an educational program for community medicine involving residential homestays in several communities and continue to carry out these educational programs involving residential homestays [4]. In theory, an educational program including homestay that more closely interacts with the residents seems to be effective. However, there are no objective data or research results that show its effectiveness.

In the future, it is necessary to examine the acceptance of homestay programs among medical students, in order to identify the pros and cons of promoting homestay programs. In addition, it is necessary to investigate whether the homestay programs can change student's attitudes toward community medicine. Thus, the present study aimed to evaluate acceptance of the homestay program and attitude toward community medicine among medical students.

## Methods

### Outline of the educational program

An educational program for community medicine (community medicine summer seminar) for medical students was held in Tamba City, a rural area in the Hyogo Prefecture, Japan, where Hyogo Prefectural Kaibara Hospital (renamed Hyogo Prefectural Tamba Medical Center in July 2019) is located. The study was conducted in August 2016, 2017, and 2018. The program included a homestay for two days and one night.

We recruited host families from local residents via public media, such as hospital newsletter and local newspaper, in June of each year. The local residents then applied for host family. We matched the medical students with the selected host family based on each wish and allergies to pets and food. Most of them were people who had finished raising their children. The average age of the host families was 61.9 ± 7.6 years. Many of them had finished raising their children. About 60% made regular visits to our hospital or visited relatives who were admitted to our hospital. The main factors related to matching are as follow: whether the host family wanted to accept male or female students, whether pets were allowed, the family composition of the host family (female students cannot be accepted in male-only houses; male students cannot be accepted in female-only houses).

Participants of the practical training were students who stayed with their host families for two days and one night. During the homestay, the students had dinner and breakfast and conversed with their host families. Moreover, they interviewed their host families about daily life, health, and medical issues. On the next day, the students participated in group discussions on

the issues raised in the interview. After that, the students gave a presentation on health and medical problems in the area along with their analysis.

In a follow-up questionnaire, we assessed families' homestay satisfaction and obtained host families' written consent to participate in the study. Regarding their homestay satisfaction, they answered one of the following: yes, no, or neither.

The purpose of this curriculum is to improve the affinity and their attitude of students for community medicine and rural areas.

## Study design

This analytical observational study used the survey research design. The respondents included medical students who participated in the educational program for community medicine involving residential homestays. The number of homestay participants was as follows: 15 participants (with 12 host families) in 2016, 12 (with 11 host families) in 2017, and 12 (with 10 host families) in 2018. Thirty-eight out of 39 participating students were regional quota students. All students participated only once for three years.

This study was approved by the Ethics Committee of Hyogo Prefectural Kaibara Hospital (approval number: Kai-Byo number 1216). Participants provided informed consent to publicize data obtained during this activity orally and via posters.

## Implementation and details of the questionnaire

The effect of the program was measured using a self-reported questionnaire with the visual analog scale (VAS; 0–100 mm) before and after the program. All students answered the "questionnaire before the program" at the start of community medicine summer seminar. In addition, all students answered the "questionnaire after the program" after group discussions the day after homestay. The questionnaire consisted of questions about age, sex, experience of homestay, and conversation during the homestay (we included an item on conversation during the homestay in order to confirm that conversations other than those related to the interview tasks were also conducted), and the following questions were answered with VAS:

1. Is it worthwhile for you to have experience in the field of community medicine?

2. Did you find the homestay enjoyable?

3. Does the homestay add educational significance to the program?

4. Is direct interaction with residents meaningful?

5. Is there a challenge to practicing community medicine?

6. Are you confident in practicing community medicine?

7. In the future, do you want to work in rural areas?

8. In the future, do you want to work in Tamba area where you stayed?

Questions 1–4 correspond to the medical students' acceptance of the homestay program. Questions 5–8 correspond to the attitude of medical students toward community medicine. Question 5 is about obstacle reduction. Question 6 is about self-efficacy, and questions 7 and 8 are about intention. Obstacle reduction, self-efficacy, and intention are factors correlated to behavioral transformation [5]. We selected these questionnaire items based on previous research on community-based medical education [2, 6].

The following is an example of the questionnaire answered using VAS:

Q1: Is it worthwhile for you to have experience in the field of community medicine?

No worthwhile Very worthwhile
VAS score 0 mm 75 mm 100 mm
The VAS score was 75.

## Data analysis

The results of the questionnaire were simply calculated, and the VAS scores were compared before and after the program. Statistical analysis was performed using paired *t*-test. Stata MP version 15 (Stata Corp, College Station, TX, USA) was used for statistical analysis. In all analyses, $P \leq 0.00625$ (= 0.05/8) was considered statistically significant. The Bonferroni correction method was used as a multiple comparison method to adjust for familywise error rate, in consideration of the fact that there are eight question items. Bonferroni correction adjusts the significance level according to the number of paired comparisons. Therefore, this eight refers to eight question items. Bonferroni correction, which is an adjustment made to P values when several dependent or independent statistical tests are being performed simultaneously on a single data set, was used.

## Results

The responses were obtained from the 39 medical students who participated in the program (response rate: 100%). The mean age of the participants was 21.2 ± 2.2 years (first- to sixth-year medical students). Of the participants, 24 (61.5%) were men, and 15 (38.4%) were women. Twenty-five (64.1%) students had no previous experience in homestay. None of the students had previous experience in homestay-type educational program for community medicine.

The topics during conversations with the host families, excluding those about daily life, health, and medical issues (topics of the interview as part of the program), are shown in Table 1; this information about conversations was included in the questionnaire section. The results confirmed that conversations on topics other than those related to practical training were also made. Participants conversed with local residents about everyday life, health, and medical issues during the homestay.

Table 2 depicts the changes in the acceptance of the homestay program among the medical students before and after the program. The VAS scores for all questions significantly increased. In addition, the VAS scores after the program were extremely high. In particular, the following questions had a significant increase in VAS score: "Did you find the homestay enjoyable" (before the program: 57.7 ± 20.0; after the program: 90.3 ± 9.4; differences in the scores before and after the program [95% confidence interval]: 32.6 [27.5–37.7]) and "Does the homestay add educational significance to the program" (before the program: 61.7 ± 19.4; after the program: 84.5 ± 28.6; differences in the scores before and after the program [95% confidence interval]: 22.8 [17.0 to 28.6]).

**Table 1.  Topics of the conversations between the medical students and local residents during the homestays.**

| Topics | Number of students (%) |
|---|---|
| Medical care in the area | 32 (82.1) |
| Studying | 24 (61.5) |
| Expectations about students | 24 (61.5) |
| Expectations about local hospitals | 20 (51.2) |
| History of the area | 17 (43.6) |
| Health consultation | 11 (28.2) |

**Table 2. Changes in the acceptance of the homestay program among the medical students before and after the program.**

| Question item | VAS scores before the program (mean ± SD) | VAS scores after the program (mean ± SD) | Difference (Post−Pre) 95% CI | P value |
|---|---|---|---|---|
| Is it worthwhile for you to have experience in the field of community medicine? | 78.1 ± 14.4 | 83.9 ± 13.9 | 5.8 (2.3–9.4) | 0.002* |
| Did you find the homestay enjoyable? | 57.7 ± 20.0 | 90.3 ± 9.4 | 32.6 (27.5–37.7) | < 0.0001* |
| Does the homestay add educational significance to the program? | 61.7 ± 19.4 | 84.5 ± 13.4 | 22.8 (17.0–28.6) | < 0.0001* |
| Is direct interaction with residents meaningful? | 79.9 ± 13.2 | 87.9 ± 9.6 | 8.1 (5.0–11.1) | < 0.0001* |

*significance level: P ≤ 0.00625 (= 0.05/8); Bonferroni correction.

SD, standard deviation; Difference (Post−Pre), Difference in the scores before and after the program; CI, confidence interval.

Table 3 shows the changes in the attitude toward community medicine among medical students before and after the program. The VAS scores for the following questions significantly increased: "Is there a challenge to practicing community medicine" (before the program: 70.9 ± 11.8; after the program: 76.8 ± 12.8; differences in the scores before and after the program [95% confidence interval]: 6.0 [1.8–10.2]) and "In the future, do you want to work in Tamba area where you stayed" (before the program: 56.9 ± 11.7; after the program: 71.3 ± 19.9; differences in the scores before and after the program [95% confidence interval]: 15.2 [10.6–19.9]).

## Discussion

The current study evaluated acceptance of the homestay program and attitude toward community medicine among medical students. To the best of our knowledge, this study first evaluated the acceptance of an educational program for community medicine involving residential homestays among medical students. The participants were extremely enthusiastic about the educational program for community medicine involving homestay. Moreover, the program improved the students' attitudes toward practicing community medicine.

As shown in Table 2, the medical students were extremely enthusiastic about the educational program for community medicine involving homestay. Thus, the homestay program was considered sufficiently acceptable by the students. In community-based medical education, opportunities for medical students to interact with local residents are important [2, 3]. In

**Table 3. Changes in the attitude toward community medicine among medical students before and after the program.**

| Question items | VAS scores before the program (mean ± SD) | VAS scores after the program (mean ± SD) | Difference (Post−Pre) 95% CI | P value |
|---|---|---|---|---|
| Is there a challenge to practicing community medicine? | 70.9 ± 11.8 | 76.9 ± 12.8 | 6.0 (1.8–10.2) | 0.006* |
| Are you confident in practicing community medicine? | 54.4 ± 16.5 | 60.7 ± 16.6 | 6.4 (1.5–11.2) | 0.0112 |
| In the future, do you want to work in rural areas? | 58.2 ± 13.9 | 63.8 ± 16.0 | 5.7 (1.1–10.3) | 0.0169 |
| In the future, do you want to work in Tamba area where you stayed? | 56.0 ± 11.7 | 71.3 ± 13.2 | 15.2 (10.6–19.9) | < 0.0001* |

*significance level: P ≤ 0.00625 (= 0.05/8); Bonferroni correction.

SD, standard deviation; Difference (Post−Pre), Difference in the scores before and after the program; CI, confidence interval.

the homestay program, the medical students can influence not only medical care but also the lives of residents for a longer time in close living space. These factors are significant features of homestay. All host families later expressed a favorable impression of the homestay program based on follow-up questionnaires. We believe that opportunities for medical students to closely interact with local residents are an important factor to get high practical effects and students' acceptance.

As shown in Table 3, the educational program for community medicine involving residential homestays made the practice of community medicine more interesting. In addition, the students appreciated the fact that their educational training sites could become their workplaces in the future. The program had a positive effect on barrier reduction and behavioral change [5]. General/family medicine as the first career option was strongly correlated to opting for practice in a rural area among medical students [7]. In addition, the presence of a role model and memorable experience at a class or clinical rotation were strongly correlated to opting to practice in a rural area [8–10]. A study [11] has shown that short-term exposure programs in rural areas are not effective in shaping career choices and decision-making about the location of internship. By contrast, one study has revealed that such programs are effective [6]. Our results showed that even short-term practical exposure could affect one's declared interest in future workplaces. Although the homestay program was conducted in a short period of time, students were able to spend quality time with local residents. Moreover, the educational program for community medicine involving residential homestays helped students to have in-depth conversations with local residents on various topics. Even with short-term practical exposure, this improved students' attitudes toward the practice of medicine in rural areas.

In community-based medical education, opportunities for medical students to interact with local residents are important [2, 3]. Homestay has the advantage of providing medical students the opportunity to interact more closely with local residents while dining and conversing. However, there are issues in securing host families proportionate to the number of students and ensuring safety during student homestay. A relatively small number of students participated in this homestay program. Therefore, we were able to arrange for host families through open recruitment. In order to ensure the safety of students, we interviewed each host family in advance to check the location of the house, secured the keys of the accommodation room, and ensured that there were no personality-related problems with the host family. Once again, the students were informed that if they had a problem, they should call us immediately.

The present study has several limitations. First, this study examined the short-term effects of the program on students' attitude. However, the long-term effects of the program were not fully elucidated. The positive response could be mainly attributed to students' momentary enthusiasm after a field trip. Despite medical students' acceptance of and positive attitude towards the homestay program in the short-term, follow-up for long-term outcomes will facilitate a more accurate assessment of the program. Second, almost all participating students were regional quota students. They may be more empathetic and conscious of community medicine and be inclined to work in rural areas than general medical students. In addition, only a limited number of target students could be recruited due to various reasons, such as securing host families, ensuring the safety of homestay students, and lack of prior studies showing effectiveness. Third, the host families were recruited via public media; thus, they were interested in community medicine and were highly motivated for the education of healthcare professionals in the area. Therefore, the positive change in the students' attitude may be attributed to the remarkable motivation of the host families rather than the homestay program itself.

## Conclusion

The medical students were extremely enthusiastic about the educational program for community medicine involving residential homestays. The program improved the students' attitudes toward practicing community medicine. Moreover, the students appreciated the fact that their training sites could become their workplaces in the future.

## Author Contributions

**Conceptualization:** Tsuneaki Kenzaka, Masanobu Okayama.

**Data curation:** Ken Goda, Ayako Kumabe.

**Formal analysis:** Shinsuke Yahata.

**Project administration:** Tsuneaki Kenzaka.

**Supervision:** Hozuka Akita, Masanobu Okayama.

**Writing – original draft:** Tsuneaki Kenzaka.

**Writing – review & editing:** Shinsuke Yahata, Ken Goda, Ayako Kumabe, Hozuka Akita, Masanobu Okayama.

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
