## [Decision Letter · Decision Letter 0]

10 Jul 2020

PONE-D-20-04945

Acceptance of a homestay program and attitude toward community medicine among medical students

PLOS ONE

Dear Dr. Kenzaka,

Thank you for submitting your manuscript to PLOS ONE. After careful consideration, we feel that it has merit but does not fully meet PLOS ONE’s publication criteria as it currently stands. Therefore, we invite you to submit a revised version of the manuscript that addresses the points raised during the review process.

We look forward to receiving your revised manuscript.

Kind regards,

Vijayaprasad Gopichandran

Academic Editor

PLOS ONE

Reviewers' comments:

Reviewer's Responses to Questions

**Comments to the Author**

1. Is the manuscript technically sound, and do the data support the conclusions?

Reviewer #1: Partly

Reviewer #2: No

2. Has the statistical analysis been performed appropriately and rigorously? 

Reviewer #1: Yes

Reviewer #2: Yes

3. Have the authors made all data underlying the findings in their manuscript fully available?

Reviewer #1: Yes

Reviewer #2: No

4. Is the manuscript presented in an intelligible fashion and written in standard English?

Reviewer #1: Yes

Reviewer #2: Yes

5. Review Comments to the Author

Reviewer #1: The manuscript adresses a very relevant topic in the field of undergraduate medical education. Nevertheless, it is only the report of an educational intervention with a pre-post survey (Kirkpatrick level 1) on a small sample of students, which limits its value and the possibility to generalize its results.

I suggest the following changes to the manuscript, in order to increase its methodological rigor, interest and usefulness.

- Background: the Background should give more elements about the gap in knowledge that the study aims to fill. At the end of the section, an explicit statement about the research question should be written.

- Study design and Results:

1. what are the specific objectives for this educational activities, also with respect to the genral objectives/competences of medical curriculum?

2. was there any specific mandate for students? questions to be answered? assignements to be reported after the experience?

3. constructs: the authors list "acceptance" and "attitudes" as constructs underlying the questionnaire (pag. 7, lines 106-110). How was the questionnaire designed? Was it supported by any theoretical model? Was it validated?

4. Statistics: why the alpha error was set so low (< 0.0065) instead of the usual <0.05? Why do the authors quote Bonferroni correction, while the comparison is only between paired pre-post values?

5. Table 1: the table reports a classification of the topics of conversation between the medical students and local residents. This means that the study is a mixed-method study, hence more methodological information are needed. How were these qualitative data acquired? Focus group with recording and analysis of verbatim? Which type of qualitative analysis? mechanisms to ensure trustworthiness?

6. Meals: why was this information important? It deserves a mention in the discussion, it could be a culturally mediated element.

- Discussion: I think the reader of the manuscript would like to know what the homestay specifically adds, with respect to the many other community-based educational programs and how this statement is supported by the experimental data. A wider discussion of pros and cons of these kind of programs would be a worth.

Some statements are not supported by data:

1. pg. 12, line 180: effective . Unless an educational outcome is defined and assessed, you cannot write that the program was "effective". The only measured effect is an increase in acceptance and attitude. Very seldom students explicitly dislike a program, but this does not imply that they learned anything.

2. pg. 12, lines 181-183: at the end of the sentence "These factors are the significant advantages of homestay". These factors are the EXPECTED advantages of the homestay, but you cannot know unless you assess the advantages. Maybe "features" is a better world, because it does not imply the concept of a positive effect. What did the residents think of this activity?

3. pg. 13, line 196: "... short-term practical exposure affects decision about future workplaces". Only a follow up study could give you this information. You only measured a change in a "declared interest", according to the "attitude" construct

4. pg. 13, line 199: "helped students to develop a deep connection with local residents". How can you know they developed this deeper connection? Did the students stay in touch with their hosts? Again, this is only a guess, an expected outcome

Reviewer #2: The intervention addressed in this paper is of keen interest and it is need of the hour for reforming medical education curriculum.

My comments are: -

1.The sample size is less and generalizability is questionable

2.Study design was not clearly mentioned

3.The questionnaire seems too much superficial. There are standard questionnarres available to assess the attitude of medical students towards community/rural oriented medical education and towards primary heath care service.

4.Immediate impact of the homestay programme on medical students’ attitude has been studied but there is a need to study the sustainability of the impact. The duration of the homestay program is apparently less and its impact on long term change in attitude and behaviour is questionable. The positive response could be attributed mainly to the shortlasting enthusiasm after a field trip. So, the intervention (homestay program) should be modified in terms of duration and long-term effect of attitude (follow up and outcome assessment at multiple timepoint is desirable). It is good that these limitations were reported by the author. However, I feel that the authors could have made effort to minimize these limitations.

Thus, after reviewing the paper, I feel that it might be suitable for a regional journal rather than this journal

6. PLOS authors have the option to publish the peer review history of their article (what does this mean?). If published, this will include your full peer review and any attached files.

Reviewer #1: No

Reviewer #2: No

---

## [Author Response · Author response to Decision Letter 0]

20 Aug 2020

Point-by-point response to reviewer comments

Acceptance of a homestay program and attitude toward community medicine among medical students

Reviewers' comments:

Reviewer's Responses to Questions

Comments to the Author

1. Is the manuscript technically sound, and do the data support the conclusions?

Reviewer #1: Partly 

Response: We revised the manuscript so that it is technically sound and provided more data to support our conclusions.

Reviewer #2: No 

Response: We revised the manuscript so that it is technically sound and provided more data to support our conclusions.

2. Has the statistical analysis been performed appropriately and rigorously? 

Reviewer #1: Yes 

Response: Thank you for your assessment.

Reviewer #2: Yes

Response: Thank you for this assessment.

3. Have the authors made all data underlying the findings in their manuscript fully available?

Reviewer #1: Yes 

Response: Thank you for your assessment.

Reviewer #2: No 

All data are fully available without restriction and are provided in the manuscript.

4. Is the manuscript presented in an intelligible fashion and written in standard English?

Reviewer #1: Yes 

Response: Thank you for your assessment.

Reviewer #2: Yes 

Response: Thank you for your assessment.

5. Review Comments to the Author

Reviewer #1: The manuscript adresses a very relevant topic in the field of undergraduate medical education. Nevertheless, it is only the report of an educational intervention with a pre-post survey (Kirkpatrick level 1) on a small sample of students, which limits its value and the possibility to generalize its results.

Response:　Thanks for your comment. The number of target students was limited for various reasons such as securing host families, ensuring the safety of homestay students, and lack of prior studies showing effectiveness. We have added this statement as a limitation of the present study.

I suggest the following changes to the manuscript, in order to increase its methodological rigor, interest and usefulness.

- Background: the Background should give more elements about the gap in knowledge that the study aims to fill. At the end of the section, an explicit statement about the research question should be written.

Response: Thank you for your suggestions. We have provided some information about the gap in knowledge that the study aims to address. We have also added an explicit statement about the research question in the Background.

- Study design and Results:

1. what are the specific objectives for this educational activities, also with respect to the genral objectives/competences of medical curriculum? 

Response: Thank you for your comment. We have added specific objectives of this educational activity at the end of the “Outline of the educational program” section.

2. was there any specific mandate for students? questions to be answered? assignements to be reported after the experience? 

Response: Thank you for your questions. The students were given specific mandates. Furthermore, the students gave a presentation on health and medical problems in the area and their analysis. We have added this information in the “Outline of the educational program” section.

3. constructs: the authors list "acceptance" and "attitudes" as constructs underlying the questionnaire (pag. 7, lines 106-110). How was the questionnaire designed? Was it supported by any theoretical model? Was it validated? 

Response: Thank you for your comment. We selected questionnaire items based on previous research. We have added this information in the “Implementation and details of the questionnaire” section.

4. Statistics: why the alpha error was set so low (< 0.0065) instead of the usual <0.05? Why do the authors quote Bonferroni correction, while the comparison is only between paired pre-post values? 

Response: Bonferroni correction adjusts the significance level according to the number of paired comparisons. Therefore, this eight refers to eight question items. In the Bonferroni correction, 0.0065=0.05/8. We have explained this in the “Statistics” section.

5. Table 1: the table reports a classification of the topics of conversation between the medical students and local residents. This means that the study is a mixed-method study, hence more methodological information are needed. How were these qualitative data acquired? Focus group with recording and analysis of verbatim? Which type of qualitative analysis? mechanisms to ensure trustworthiness? 

Response: This information about conversations is included in the questionnaire section. The results confirmed that conversations on topics other than those related to practical training were also made.

We included the item on conversation during the homestay, in order to confirm that topics other than those related to the tasks were well discussed. We have described this in the “Implementation and details of the questionnaire” section of the manuscript. In addition, we confirmed that topics other than other related to practical training were also discussed during the homestay. Results of this assessment are also provided.

6. Meals: why was this information important? It deserves a mention in the discussion, it could be a culturally mediated element. 

Response: Regarding the content of meals, there was no significant difference in the results with regard to medical students’ acceptance of the homestay program. However, since this statement could be misleading, we deleted the mention of meals.

- Discussion: I think the reader of the manuscript would like to know what the homestay specifically adds, with respect to the many other community-based educational programs and how this statement is supported by the experimental data. A wider discussion of pros and cons of these kind of programs would be a worth.

Response: Thank you for this suggestion. We provided a more detailed discussion of the merits and demerits of the homestay program in the Discussion section.

Some statements are not supported by data:

1. pg. 12, line 180: effective . Unless an educational outcome is defined and assessed, you cannot write that the program was "effective". The only measured effect is an increase in acceptance and attitude. Very seldom students explicitly dislike a program, but this does not imply that they learned anything. Response: Thank you for this comment. We changed the word “effective” to “acceptable.” 

2. pg. 12, lines 181-183: at the end of the sentence "These factors are the significant advantages of homestay". These factors are the EXPECTED advantages of the homestay, but you cannot know unless you assess the advantages. Maybe "features" is a better world, because it does not imply the concept of a positive effect. What did the residents think of this activity? 

Response: Thank you for pointing this out. We changed the word “advantages” to “features”. The host family had a favorable impression of the homestay program. We added these. However, detailed results of this will be provided in another paper.

3. pg. 13, line 196: "... short-term practical exposure affects decision about future workplaces". Only a follow up study could give you this information. You only measured a change in a "declared interest", according to the "attitude" construct. Response: Thank you for this comment. We changed the word “decision” to “declared interest.” 

4. pg. 13, line 199: "helped students to develop a deep connection with local residents". How can you know they developed this deeper connection? Did the students stay in touch with their hosts? Again, this is only a guess, an expected outcome. Response: Thank you for this comment. We changed the word “connection” to “conversations.” We added the following sentence: “Moreover, the educational program for community medicine involving residential homestays helped students to have in-depth conversations with local residents on various topics.”

Reviewer #2: The intervention addressed in this paper is of keen interest and it is need of the hour for reforming medical education curriculum.　　　　　　　　　　　　　　　　　　　　　　　　　　　　　　　　　 Response: Thank you for your comment.

My comments are: -

1.The sample size is less and generalizability is questionable Response: Thank you for your comment. As pointed out, the sample size is small. This has been discussed in the Limitations section as follows: “only a limited number of target students could be recruited due to various reasons, such as securing host families, ensuring the safety of homestay students, and lack of prior studies showing effectiveness.” Moreover, as a multiple comparison method for adjusting the familywise error rate, the Bonferroni correction method was used in consideration of the fact that there are eight question items. This makes the study more rigorous and is expected to compensate for the small sample size.

2.Study design was not clearly mentioned. 

Response: Thank you for your comment. We have specified the study design in the manuscript.

3.The questionnaire seems too much superficial. There are standard questionnarres available to assess the attitude of medical students towards community/rural oriented medical education and towards primary heath care service. Response: Thank you for this comment. We selected these questionnaire items based on previous research. We added this information in the “Implementation and details of the questionnaire”.

4.Immediate impact of the homestay programme on medical students’ attitude has been studied but there is a need to study the sustainability of the impact. The duration of the homestay program is apparently less and its impact on long term change in attitude and behaviour is questionable. The positive response could be attributed mainly to the shortlasting enthusiasm after a field trip. So, the intervention (homestay program) should be modified in terms of duration and long-term effect of attitude (follow up and outcome assessment at multiple timepoint is desirable). It is good that these limitations were reported by the author. However, I feel that the authors could have made effort to minimize these limitations. 

Response: Thank you for this suggestion. We have added this point in the Limitations section. We obtained information on medical students' acceptance of the homestay program. Furthermore, we assessed students’ attitude toward community medicine for short-term. Long-term outcomes must be followed up in future study. 

Thus, after reviewing the paper, I feel that it might be suitable for a regional journal rather than this journal. 

Response: Thank you for this comment. However, there are no previous objective data or research results that report the effectiveness about educational program involving residential homestays. Therefore, we feel sure that this new impact of the homestay program is suitable for PLOS ONE.

Further to the email just sent to you, please see below for an additional journal request;

In the methods, please provide additional details about ho the residents applied to be host families, and describe how medical students were matched to families. Additionally, please clarify if host families consented to participate in research and indicate how consent was recorded.

Response: Thank you for this comment. We have added additional information. In addition, we assessed students’ homestay satisfaction and obtained host families’ written consent to participate in the study in a follow-up questionnaire. We have added these in the “Outline of the educational program” section.

6. PLOS authors have the option to publish the peer review history of their article (what does this mean?). If published, this will include your full peer review and any attached files.

Response: We choose “no.”

---

## [Editor Report · Decision Letter 1]

25 Aug 2020

Acceptance of a homestay program and attitude toward community medicine among medical students

PONE-D-20-04945R1

Dear Dr. Kenzaka,

We’re pleased to inform you that your manuscript has been judged scientifically suitable for publication and will be formally accepted for publication once it meets all outstanding technical requirements.

Kind regards,

Vijayaprasad Gopichandran

Academic Editor

PLOS ONE
---

## [Editor Report · Acceptance letter]

28 Aug 2020

PONE-D-20-04945R1 

Acceptance of a homestay program and attitude toward community medicine among medical students 

Dear Dr. Kenzaka:

I'm pleased to inform you that your manuscript has been deemed suitable for publication in PLOS ONE. Congratulations! Your manuscript is now with our production department. 

Kind regards, 

on behalf of

Dr. Vijayaprasad Gopichandran 

Academic Editor

PLOS ONE